# Subgroup-Specific Diagnostic, Prognostic, and Predictive Markers Influencing Pediatric Medulloblastoma Treatment

**DOI:** 10.3390/diagnostics12010061

**Published:** 2021-12-28

**Authors:** Sutapa Ray, Nagendra K. Chaturvedi, Kishor K. Bhakat, Angie Rizzino, Sidharth Mahapatra

**Affiliations:** 1Department of Pediatrics, University of Nebraska Medical Center, 601 S Saddle Creek Road, Omaha, NE 68198, USA; sutapa.ray@unmc.edu (S.R.); nchaturvedi@unmc.edu (N.K.C.); 2Fred and Pamela Buffet Cancer Center, University of Nebraska Medical Center, Omaha, NE 68105, USA; kishor.bhakat@unmc.edu (K.K.B.); arizzino@unmc.edu (A.R.); 3Department of Genetics, Cell Biology, and Anatomy, University of Nebraska Medical Center, Omaha, NE 68198, USA; 4Eppley Institute for Research in Cancer and Allied Disease, University of Nebraska Medical Center, Omaha, NE 68198, USA; 5Department of Biochemistry and Molecular Biology, University of Nebraska Medical Center, Omaha, NE 68198, USA

**Keywords:** medulloblastoma, WNT, SHH, group 3, group 4

## Abstract

Medulloblastoma (MB) is the most common malignant central nervous system tumor in pediatric patients. Mainstay of therapy remains surgical resection followed by craniospinal radiation and chemotherapy, although limitations to this therapy are applied in the youngest patients. Clinically, tumors are divided into average and high-risk status on the basis of age, metastasis at diagnosis, and extent of surgical resection. However, technological advances in high-throughput screening have facilitated the analysis of large transcriptomic datasets that have been used to generate the current classification system, dividing patients into four primary subgroups, i.e., WNT (wingless), SHH (sonic hedgehog), and the non-SHH/WNT subgroups 3 and 4. Each subgroup can further be subdivided on the basis of a combination of cytogenetic and epigenetic events, some in distinct signaling pathways, that activate specific phenotypes impacting patient prognosis. Here, we delve deeper into the genetic basis for each subgroup by reviewing the extent of cytogenetic events in key genes that trigger neoplastic transformation or that exhibit oncogenic properties. Each of these discussions is further centered on how these genetic aberrations can be exploited to generate novel targeted therapeutics for each subgroup along with a discussion on challenges that are currently faced in generating said therapies. Our future hope is that through better understanding of subgroup-specific cytogenetic events, the field may improve diagnosis, prognosis, and treatment to improve overall quality of life for these patients.

## 1. Introduction

Medulloblastoma (MB), a primitive neuroectodermal tumor (PNET), is concurrently the most common malignant pediatric brain tumor and the leading cause of cancer-related childhood mortality [1,2,3,4,5]. Annual incidence for pediatric medulloblastoma in the United States is approximately 500 cases [1,6,7]. Accounting for 40% of tumors arising in the posterior fossa, these tumors can grow rapidly and invade important structures, triggering cerebellar dysfunction and disrupting cerebrospinal fluid circulation (Figure 1). Clinical presentation and symptomatology can reflect this pattern of growth. Children often present with difficulties in coordination and gait (cerebellar signs) and/or with early morning headaches, nausea, vomiting, papilledema, and double vision (hydrocephalus); average time from onset of symptoms to diagnosis can be as short as 2–3 months [8,9].

Genome-wide high-throughput analyses of large cohorts of medulloblastoma patients have revealed four distinct molecular subgroups, each possessing unique genetic and epigenetic alterations [10,11,12,13,14,15,16,17,18,19,20]. Medulloblastomas are now subdivided into wingless (WNT), sonic hedgehog (SHH), group 3, and group 4 tumors, with each subgroup possessing a distinct transcriptomic and methylation profiles, somatic genetic aberrations, demographic distributions, histologies, and clinical outcomes (Figure 2) [7,21]. However, these primary subgroups can be further subdivided into at least 12 subtypes on the basis of clustering of unique molecular and clinical features influencing overall risk and survivability (Table 1) [22].

The current mainstay for management of medulloblastoma starts with surgical resection. The intensity of post-surgical treatment, i.e., with craniospinal irradiation (CSI) and adjuvant chemotherapy, is determined by factors that divide patients into average- and high-risk treatment groups (Table 2) [23]. Average-risk patients (defined as children older than 3 years of age with near total resection of tumor and the absence of metastatic disease) have an expected 5-year overall survival of ~85% [24,25,26]. They are typically treated with 23.4 Gy craniospinal irradiation followed by adjuvant chemotherapy with cisplatin, vincristine, and either cyclophosphamide or lomustine [25]. Patients with high-risk tumors (arising in children younger than 3 years of age with less than near total tumor resection and/or metastatic disease at presentation) have a survival closer to 60–70% [26,27,28]. They typically receive higher CSI doses of 36–39 Gy, followed by adjuvant chemotherapy with cisplatin, vincristine, and cyclophosphamide [24,29,30]. However, recent studies have highlighted the need for subgroup-specific treatment stratification. For example, in group 3 tumors alone, the addition of carboplatin to adjunctive chemotherapy dramatically improved overall survival from 54% to 73% [29].

Similarly, infants represent a unique group of high-risk patients that have necessitated delayed radiation therapy and treatment with multi-agent chemotherapy followed by autologous hematopoietic stem cell rescue and methotrexate [31,32,33]. Radiation dose reduction has the important benefit of mitigating treatment-related neurocognitive deficits [34]. However, this approach has provided better outcomes for children with gross total resection with an absence of metastatic dissemination compared to patients with residual or metastatic disease [35,36].

In this review, we discuss medulloblastoma using international genome-based studies to explore mutations within critical signaling pathways that result in the unique protein profile signatures distinguishing each subgroup [11,12,13,14,15,16,17,18,19,20]. We provide subgroup-specific demographic and prognostic data with a discussion on the potential diagnostic and predictive markers that may be used for both prognostication and treatment purposes. These include a review of changes in the expression of non-coding RNAs that have been found to associate with MB. We further introduce potential new therapeutic options derived from thorough exploration of the molecular basis for each subgroup, including a brief overview of anti-neoplastic therapies currently in pre-clinical and/or clinical trials. 

## 2. Wingless (WNT) Medulloblastoma

The WNT subgroup characterizes approximately 10% of medulloblastoma cases [13,37,38]. These tumors typically arise in the midline, away from the cerebellum in the dorsal brainstem from progenitor cells originating in the lower rhombic lip during cerebellar development (Figure 1) [39]. Peak age of onset is between 10 and 12 years with equal incidence in males and females; it is rarely ever encountered in infants [13]. Patients with this subgroup are distinguished from all other MB subtypes given their classic histology and distinctly favorable prognosis, with a 5-year overall survival exceeding 95% (Figure 2) [7,13,37,40]. Even patients presenting with metastasis in 5–10% of cases experience good prognosis (Table 1) [22]. 

The Wingless (WNT) signaling pathway is highly conserved and critical to normal neurologic development [41,42]. Via the canonical pathway, WNT binds to its primary receptor complex, Frizzled (Fz) and LRP5/6, which recruits and phosphorylates Dishevelled (DVL), in turn leading to inactivation of a multi-protein complex (the β-catenin destruction complex) responsible for β-catenin degradation [41,43,44]. β-Catenin is then free to translocate to the nucleus to interact with T-cell factor/lymphoid enhancer factor-1 (TCF/LEF-1) transcription factors, which activate cell proliferation and survival genes, such as *c-Myc* and *cyclin D1* [5,41,44]. In the absence of WNT, the destruction complex, consisting of AXIN1, adenomatous polyposis coli (APC), casein kinase 1α (CK1α), and glycogen synthase kinase 3β (GSK3β), inactivates β-catenin via phosphorylation at key serine residues, leading to subsequent polyubiquitination and degradation [41,45]. In most sporadic forms of WNT medulloblastoma, a high frequency of mutations have been identified in the proto-oncogene encoding β-catenin on chromosome 6, i.e., *CTNNB1* [11,46]. *These mutations cluster around key phosphorylation sites that render* β-catenin resistant to degradation, thus leading to constitutive activation of proliferative nuclear signals and subsequent neoplastic transformation [5,47,48,49]. 

### 2.1. Targeting WNT/β-Catenin Signaling 

Given the WNT subgroup possesses the best prognostic profile amongst medulloblastoma patients, targeted therapeutics that have shown promise in pre-clinical studies have yet to find their way to clinical trials. The first identified inhibitor was a derivative of the non-steroidal anti-inflammatory drug, celecoxib, i.e., OSU-03012 (AR-12), originally designed as a PDK1 inhibitor. By specifically inhibiting GSK-3β phosphorylation and inactivation, AR-12 triggered apoptosis and cell cycle arrest in D283 and D324 cells. Oral supplementation of AR-12 to D283 xenograft mice resulted in significantly reduced tumor volume and reduced WNT target proteins, β-catenin, c-Myc, and cyclin D1 levels [50]. Next, ginkgetin was identified, a natural bioflavonoid isolated from the Chinese plum-yew, which inhibited AXIN2, cyclin D1, and survivin, and induced cell cycle arrest at the G_2_/M phase in DAOY and D283 cells [51]. Similarly, in a time- and dose-dependent manner, curcumin inhibited growth of DAOY cells [52]. Finally, norcantharidin, a natural terpenoid secreted by blister beetles, demonstrated anti-neoplastic properties in medulloblastoma cells by inhibiting phosphatase 2A (PP2A), which is involved in β-catenin stability [53]. Moreover, norcanthardin also inhibited nuclear β-catenin translocation, induced G_2_ cell cycle arrest, and activated apoptosis in vitro and in vivo [54]. 

As opposed to other subgroups with more unfavorable survival profiles, focus for WNT patients has instead shifted to limiting treatment-related neurocognitive and endocrinologic sequelae secondary to direct craniospinal irradiation during critical years of growth, which can include growth delay, thyroid dysfunction, hyperprolactinemia, and gonadal damage [55]. Strong efforts are now underway to potentially limit cytotoxic therapies in an effort to improve post-treatment quality of life [23,56,57]. In the PNET 5 MB trial (NCT02066220), children with standard risk MB, i.e., near-total resection, β-catenin nuclear staining, and non-metastatic, are receiving reduced-intensity chemotherapy (6 cycles vs. 8 cycles) vs. concurrent carboplatin during radiotherapy [57]; in two other concurrent active trials (NCT01878617 and NCT02724579), WNT patients are being treated with reduced craniospinal irradiation + limited targeted boost to tumor bed and reduced chemotherapy (with cisplatin, vincristine, and lomustine) (Table 3) [57]. The aim of all three trials is to determine if these patients with lower risk can fare equally well with a more tailored approach to their radio-chemotherapy regimens. 

### 2.2. Therapeutically Activating WNT/β-Catenin Signaling 

Contrary to the primary held mechanism for WNT tumorigenesis via constitutive β-catenin activation, some studies have shown that an overexpression of β-catenin may actually lead to anti-neoplastic effects. For example, in transformed epidermal keratinocytes [58] and colorectal cancer cells [59], overexpression of β-catenin led to cell cycle arrest at G_2_/M and subsequent apoptosis. In MB cell lines, transient overexpression of β-catenin resulted in specific cell cycle arrest at the G_2_/M phase of ONS-76 cells (SHH-type) and a reduced in PTCH1 and SMO in UW473 cells (group 3); all subtypes of cell lines experienced a reduction in colony formation [43]. Subsequently, in an SMO activation murine model of SHH MB, WNT activation was shown to reduce cell proliferation and tumor size and prolong survival [60]. More recently, in patient-derived group 3 and 4 MB brain tumor initiating cells (BTIC), ectopic WNT activation reduced cell proliferation, self-renewal, secondary tumor sphere formation, and orthotopic tumor burden with improved xenograft survival. On the basis of single-cell RNA sequencing, the authors discovered that a subset of WNT active cells within groups 3 and 4 exhibited decreased proliferative and self-renewal properties. Interestingly, the WNT gene signature within these cells, i.e., elevated *Axin2* and reduced *Bmi1* and *Sox2*, when isolated in group 3 patients, was predictive of longer survivorship and improved overall prognosis [61]. These findings suggest a paradigm shift in potentially exploiting WNT activation in the treatment of non-WNT tumors. 

## 3. Sonic Hedgehog (SHH) Medulloblastoma

Sonic hedgehog medulloblastomas (SHH MB) comprise approximately 30% of all MB and arise in the cerebellum from granule neuron precursor cells due to constitutive activation of the sonic hedgehog signaling pathway (Figure 1) [38,39]. The five-year overall survival of SHH MB patients is ~70%. However, as discussed below, survival of these patients varies significantly due to differences in their mutation status. SHH-MB occur most often in infants (<3 years of age) and adults (>17 years of age), with a smaller fraction of cases occurring during childhood and adolescence. SHH MB is the best genetically characterized form of MB, with nearly all SHH MB having either germline or somatic mutations in genes associated with hedgehog signaling (Figure 2). 

In wild-type cells, hedgehog signaling is activated by binding of ligand (Desert, Indian, or Sonic) to the cell surface receptor Patched (PTCH). Binding of ligand to PTCH increases the expression and activity of the seven-membrane spanning receptor Smoothened (SMO) in the primary cilium where it blocks the action of Suppressor of fused (SUFU), which sequesters GLI transcription factors in the cytoplasm. As a consequence of SMO activation, GLI transcription factors translocate into the nucleus where they activate transcription of hedgehog target genes. In SHH MB, mutations occur at different points along the hedgehog signaling pathway, including deletion or loss of function mutations in *PTCH1* (~40%) or *SUFU* (~10%), gain of function (activating) mutations in *SMO* (9%), or amplification of *GLI2* (6%) [62,63]. As a result of these mutations, most SHH MB are not dependent on ligand activation of the hedgehog pathway. However, in a small percentage of childhood and adolescent SHH MB, *Sonic* is amplified. PTCH1 mutations occur in all age groups, whereas *SUFU* and *SMO* mutations are more frequent in infants and adults, respectively [63]. *GLI2* mutations occur in children, but they have not been observed in infant or adult SHH MB [63].

In addition to hedgehog signaling mutations, SHH MB exhibit a wide range of mutations, including mutations in *MYCN*, *TP53*, *TERT*, and *PTEN* [63]. Loss or silencing of *PTEN*, which activates PI3K/AKT/mTOR signaling, has been reported to drive nonconical activation of GLI transcription factors and aberrant hedgehog signaling [64,65]. *TP53* status of SHH MB is highly significant because *TP53* mutations in these tumors are highly correlated with patient outcomes. Lack of *TP53* mutations in young children and adults is associated with good outcome (5-year OS > 80%). In contrast, *TP53* mutations, which frequently occur in older children (median age of 9), are associated with poor overall survival (40% 5-year OS) [63]. Notably, *TP53* mutations are found in approximately two-thirds of children over 6 years of age who die of SHH MB, and over 50% of *TP53* mutations in SHH MB are germline mutations [66].

### 3.1. Molecular Characteristics and Subtypes of SHH MB

SHH-MB exhibit four histopathologies: classic, desmoplastic/nodular (DN), large cell/anaplastic (LC/A), and medulloblastoma with extensive nodularity (MBEN). DN and classic occur most often in SHH MB (35 to 45% for each histology). LC/A and MBEN occur at frequencies of 15 and 10%, respectively. Histology alone is insufficient for predicting patient outcome or selection of treatment options. Recently, SHH MB were further subdivided into four subtypes (α, β, γ, δ) on the basis of their gene expression profiles and DNA methylation status (Table 1) [22]. SHHα, SHHβ, SHHγ, and SHHδ were found to account for 29, 16, 21, and 34% of SHH MB, respectively. SHHα carries very high risk of dying and exhibits classic, DN, or LC/A histology. SHHβ carries high risk (lower than SHHα) and exhibits classic or DN histology. SHHγ carries low risk and exhibits classic, DN, or MBEN histology. SHHδ carries average risk and exhibits classic or DN histology [22].

SHHα, which has the worst outcome, occurs primarily in childhood and adolescence (3 to 16 years of age). This subtype exhibits *MYCN*, *GLI2*, and *YAP1* amplifications at frequencies of 20%, 16%, and 8%, respectively. *TP53* mutations occur frequently in SHHα (35%), and thus far is the only identified negative prognostic marker for SHH MB. Infant SHH MB are comprised mainly of SHHβ and SHHγ subtypes. Of the two subtypes, SHHβ exhibits poorer outcome than SHHγ, possibly due to a higher frequency of metastasis by SHHβ (33%) than SHHγ (9%). SHHβ frequently possesses *PTEN* deletions (25%) and multiple focal amplifications. SHHγ exhibits the lowest mutational load of SHH MB. SHHδ, which occur primarily if not exclusively, in adults, has favorable outcome. It often carries *TERT* promoter mutations (90%) that elevate telomerase [22].

### 3.2. Challenges in Treating SHH MB

Although the five-year overall survival of SHH MB is ~70%, treatment of many SHH MB patients, in particular infants, is very challenging. The standard of care for the treatment of SHH MB is similar to that used for all MB patients over the age of 3. For infants, radiation-sparing therapy is used to minimize the cytotoxic effects on the developing brain. Currently, there are several ongoing clinical trials for SHH MB (Table 3). In addition, a recent clinical trial (SJYC07) reported that a subset of SHH MB patients (SHH-II) according to their DNA methylation status exhibited improved progression-free survival [67]. Rather than providing a comprehensive overview of new treatment options for SHH MB, below we focus on SMO inhibitors to illustrate the challenges faced when treating SHH MB with small molecule inhibitors. Broader coverage of treatment options for patients with SHH MB has been reviewed recently [68].

Multiple studies, including several recent large-scale studies, have demonstrated that survival of all MB patients, including those with SHH MB, are highly dependent on the molecular characteristics of the tumor [22,62,63]. In the case of SHH MB, targeted therapy has focused heavily on the SMO inhibitors vismodegib and sonidegib because almost all SHH MB patients have mutations in genes directly associated with aberrant hedgehog signaling. Both SMO inhibitors have been evaluated in MB clinical trials [69,70,71]. In a meta-analysis of phase I and phase II clinical trials, the overall response of patients with SHH MB to these inhibitors was 37%, with sonidegib exhibiting at least threefold higher efficacy than vismodegib [69]. Not surprisingly, there were no positive responses observed in patients with other types of MB.

Although the clinical responses to SMO inhibitors are encouraging, the long-term efficacy of these inhibitors is limited by the development of resistance in at least 20% of those who had responded favorably to SMO inhibitors. Resistance appears to be due, at least in part, to mutations in the drug binding pocket of SMO that interfere with drug binding [72,73]. Additionally, SHH MB patients who become resistant to vismodegib are likely to exhibit cross-resistance to sonidegib [74]. As a result, efforts are underway to develop new SMO inhibitors, including those that inhibit mutant drug-resistant SMO [75,76]. The use of vismodegib and sonidegib is also contraindicated for pediatric patients because these drugs can hinder bone development and cause irreversible growth plate fusions, leading to short stature and disproportionate growth [77]. A further limitation of SMO inhibitors is expected for SHH MB patients with mutations downstream of SMO, in particular for those with SUFU loss of function mutations and those with *GLI2* amplification, which occur in 16 and 6% of SHH MB patients, respectively [73]. Moreover, recent evidence collected from the study of genetically engineered mouse (GEM) models of SHH MB has led to the suggestion that treatment with vismodegib can epigenetically silence SUFU and drive tumor cell proliferation [78].

Another challenge in the treatment of SHH MB with small molecule inhibitors is tumor heterogeneity. In tumors with significant heterogeneity, which is the case for most tumors, the responses of the different tumor cell populations can vary widely. This is especially true of the tumor-initiating cell population of the tumor, which is often less sensitive to the inhibitor than the bulk of the tumor. This problem was laid bare in the study of a GEM model of SHH MB. In this model, irradiation of *PTCH*^+/-^ mice resulted in the vast majority of the mice (~80%) developing tumors that closely resembled human SHH MB, enriched in the stem cell transcription factor SOX2 [79,80]. Analysis of the tumors from this GEM model indicated that the tumors were hierarchically organized with a small population (<5%) of SOX2-positive (SOX2^+^) cancer stem cells that self-renewed, yet proliferated infrequently [79]. This rare, label-retaining SOX2^+^ stem cell population gave rise to Dcn^+^ progenitor cells, which proliferated actively and gave rise to non-proliferating/post-mitotic NeuN^+^ differentiated cells. Collectively, the proliferative progenitor population and their differentiated derivatives accounted for the bulk of the tumor. Treatment of these tumor-bearing mice with vismodegib or cytarabine (Ara C) targeted the proliferating population and spared the infrequently proliferating SOX2^+^ cell population. As a result, while the tumor did regress, upon drug discontinuation, tumors regrew [79]. The critical role of the SOX2^+^ cell population in these tumors was further demonstrated with the finding that SOX2^+^ cells exhibited substantially higher tumor-initiating capacity than SOX2^−^ cells. Moreover, lineage-tracing studies have directly implicated a rare SOX2^+^ granule neuron precursor population as the cell of origin of SHH MB [81], and MB patients with high levels of SOX2 have poorer overall survival [79].

### 3.3. Meeting the Challenges Posed by SOX2^+^ SHH MB Cancer Stem Cells

Meeting the major challenge of tumor recurrence for SHH MB will require targeting the stem cell population of SOX2^+^ cells. One approach to addressing this daunting challenge is to identify the vulnerabilities of the population of infrequently proliferating SOX2^+^ tumor cells. Normal SOX2^+^ stem cells are present in many developing tissues [82,83]. SOX2^+^ stem cells during development give rise to highly proliferative progenitor cell populations, which then generate tissue-specific differentiated cells that constitute the bulk of the developing tissue [82,83]. Importantly, elevating SOX2 in the progenitor population during development reduces their proliferation, whereas interfering with SOX2 function in the stem cell compartment enhances proliferation. These studies indicate that SOX2 levels are used by both normal stem cells and cancer stem cell populations to control proliferation [84].

Our recent studies have shown that elevating SOX2 from an inducible promoter in a wide range of human tumor cell lines, including the SHH MB cell lines, can lead to a dramatic reduction of proliferation both in vitro and in vivo [85,86,87,88]. In a flank tumor model using ONS76 MB cells, elevation of SOX2 from an inducible promoter led to a reversible state of tumor growth arrest [88]. Upon removal of the inducer, SOX2 levels returned to baseline and tumor growth resumed to match their uninduced counterparts. This reversible SOX2 growth-arrested tumor model system parallels the behavior of the rarely proliferating SOX2^+^ SHH MB cancer cells discussed above. 

This SOX2-inducible model system could be exploited to determine the molecular mechanism by which elevation of SOX2 halts tumor growth. Insights into the mechanisms that limit growth of the tumor could help develop clinical strategies that prevent the SOX2^+^ SHH MB tumor-initiating cells from repopulating the tumor after completion of standard of care. Second, this SOX2-inducible model system could be used as a platform to screen a library of drugs, particularly FDA-approved drugs, for their ability to eradicate SOX2 growth-arrested tumor cells in vivo. Progress on either of these experimental approaches could help develop novel strategies to either delay or prevent SHH MB recurrence, which occurs with an incidence of ~25% and is nearly always fatal [89,90]. 

## 4. Group 3 Medulloblastoma

Group 3 tumors represent roughly 25% of all MB cases and arise from nestin-positive neural stem cells in the cerebellar vermis (Figure 1) [38,91,92]. This subgroup shows a male predominance (male to female ratio 2:1) and occurs almost exclusively in infants and young children, with the highest rates of diagnosis occurring between ages 3 and 5 years [22,41,42]. Group 3 MBs mostly demonstrate either classic or LCA histology and frequently present with high rates of metastasis at diagnosis (up to 50% of patients). These high risk factors, i.e., young age, metastatic spread at diagnosis, presence of *MYC* amplifications, and prevalent LCA histology, all contribute to these tumors having the worst prognosis of any subgroup with an overall 5-year survival of <50%, especially in MYC-amplified tumors (Figure 2) [3,41,93,94].

### 4.1. Molecular Characteristics and Subtypes of Group 3 MB

The biology of group 3 tumors is not well established. These tumors likely originate from a neural/cerebellar stem cell population [91,95]. Unlike WNT/SHH-driven MBs, group 3 tumors are genetically heterogenous, and extensive integrated genomic and molecular analyses have yet to identify a common driver pathway that defines the group 3 subgroup. Nevertheless, somatic *MYC* amplification (17%) is most frequently observed, often concurrent with genomic rearrangement of *PVT1–MYC* fusion [16]. Although recurrent somatic mutations are rarely encountered, genetic mutations in four genes, *SMARC4*, *KBTBD4*, *CTDNEP1*, and *KMT2D*, have been identified in over 5% of cases [62]. Small subsets of these tumors are driven by amplifications of *MYCN* (5%) and the transcription factor *OTX2* (3%) [62]. Enhancer hijacking events leading to the activation of GFI1/GFI1B are present in 15–20% of group 3 MBs and may also be important driver events for this subgroup [96]. Other genetic aberrations include copy number alterations in TGFβ pathway genes and mutations in *NOTCH* signaling genes [97]. Finally, cytogenetic alterations are abundantly seen in group 3 tumors. The most common cytogenetic abnormalities in this subgroup include the gain of 17q (58%) and loss of 17p (55%), along with a loss of 16q (42%), 10q (43%), and 9q (21%) and gain of 7 (39%) and 1q (41%) [41,98].

Recently, two newer molecular classifications have been proposed for group 3 MB. One classification is based on methylome data that distinguishes a high-risk subtype with frequent *MYC* amplifications [99]. The second defines three primary subtypes: 3α (occurring in infants and young children with frequent metastasis), 3β (occurring in older children displaying *OTX2* gain, *DDX31* loss, and high *GFI1/GFI1B* amplification/activation), and 3γ (occurring in infants with *MYC* amplification) (Table 1) [22].

### 4.2. Risk Stratification, Prognostic Biomarkers, and Treatment Approaches

A recent consensus proposal recommended a more refined risk stratification for MB tumors in pediatric patients. This consensus proposed four categories: low (>90% survival), standard (75–90% survival), high (50–75% survival), and very high risk (<50% survival) MBs [100]. The non-metastatic group 3 MBs with no *MYC* amplification are included in standard-risk population. However, the metastatic group 3 MBs with MYC amplification are included in very high-risk stratification. The risk identification of non-metastatic group 3 with *MYC* amplification or LCA histology or isochromosome 17q needs to be further clarified. 

Our limited understanding of tumorigenesis mechanisms for group 3 tumors has arrested the development of targeted treatment strategies. However, currently, for newly diagnosed patients, the NCT01878617 trial offers a risk-adapted treatment strategy: intermediate- and high-risk patients are exposed to new chemotherapy agents (pemetrexed and gemcitabine) after standard chemotherapy and risk-adapted radiotherapy. For recurrent and refractory MB patients, two trials offer treatments based on molecular subgrouping. In NCT04023669, prexasertib (LY2606368), a targeted checkpoint kinase 1/2 (CHK1/2) inhibitor, is administered in combination with cyclophosphamide or in combination with gemcitabine for group 3 MBs. The trial NCT03434262 investigates the combination of the cyclin-dependent kinase inhibitor ribociclib and gemcitabine for group 3 MBs (Table 3) [101].

The use of alternative adjuvant chemotherapy for high-risk MB is being actively pursued. In a recent prospective randomized phase 3 clinical trial (NCT00392327), therapy intensification with carboplatin as a radiosensitizer improved event-free survival from 54% to 73% at 5 years for children with high-risk group 3 MB. On the basis of these results, concurrent treatment of carboplatin with radiotherapy is recommended for children with high-risk group 3 MB [29].

Immunotherapy has recently gained attention as a novel adjunctive therapy for recurrent, progressive, or refractory MB tumors. The challenge has always been overcoming the intricate system of barriers that complicates delivery and efficacy [102]. However, several immunotherapeutic approaches have reached clinical trials for MB, particularly immune checkpoint inhibitors that target the immunosuppressive brain tumor microenvironment [103]. To date, the most notable examples of effective immune checkpoint inhibitors are antibodies against cytotoxic-T-lymphocyte antigen 4 (CTLA-4) and the programmed cell death protein (PD-1) and its ligand PD-L1. Currently, there are three active phase 1/2 clinical trials that aim to evaluate the use of PD-1/PD-L1 inhibitors in MB: pembrolizumab (NCT02359565), nivolumab (NCT03173950), and nivolumab in combination with the CTLA-4 inhibitor ipilimumab (NCT03130959) (Table 3).

### 4.3. Pre-Clinical Group 3 Models Identify Druggable Pathways 

Our lack of clear understanding on the pathophysiology triggering group 3 tumors has also hampered the generation of spontaneous animal models for this subgroup. Given the high frequency of *MYC* mutations and their influence on prognosis in group 3 MB, models of MYC-driven tumors have emerged. While *MYC* overexpression alone is insufficient in triggering tumors, combining it with inactivating mutations of the tumor suppressor *TP53* results in tumors exhibiting an LCA histology with similarity in gene expression signatures to group 3 tumors [96,104,105,106]. The generated tumors are enriched for genes targeting PI3K/mTOR. Drug screening within this model identified histone deacetylase inhibitors (HDACIs, such as panobinostat) demonstrating synergistic activity with phosphatidylinositol 3-kinase inhibitors (PI3KI) via activating the expression of the FOXO1 tumor suppressor [107]. Another mouse model utilizing human neural stem and progenitor cells harboring transformed *MYC*, dominant-negative *TP53*, and constitutively active *AKT* and *hTERT* revealed tumor sensitivity to cyclin-dependent kinase (CDK) inhibitors, such as Palbociclib [108]. These studies not only highlight the critical role MYC played but also revealed novel downstream therapeutic targets. 

However, *TP53* mutations are rarely detected in group 3 tumors at diagnosis [66]. Thus, such mouse models that require *TP53* mutation may be of limited relevance for understanding human tumor biology and future therapy development. A recent study by Tao et al. reported that SOX2^+^ cells of the early postnatal cerebellum can be transformed by MYC alone, resulting in tumors resembling human group 3 MB based on histology and gene-expression profiling. Using this model, they also identified lactate dehydrogenase A (LDHA) as a novel therapeutic target for group 3 MB [109]. A separate study overexpressed MYC and GFI1 in CD133+ cells triggering group 3β tumors [96].

Despite the central role that MYC plays in high-risk group 3 tumors, MYC has proven thus far to be therapeutically undruggable because of its complex structure and non-enzymatic/pleotropic nature. However, targeting epigenetic regulators of MYC may provide a promising alternative. Bromodomain and extra-terminal (BET)-containing proteins promote gene transcription by recognizing side chain acetylated lysine on open chromatin and have been shown as potential targets of MYC or MYCN transcription [110]. BET bromodomain inhibitors, such as JQ1, inhibited in vitro cell proliferation and prolonged survival in MYC-amplified MB xenografts [111]. Recent preclinical studies in cell lines and xenograft models of group 3 MB also identified that BET protein inhibitors not only had single agent antitumor efficacies, but also synergistic efficacies with PI3K/mTOR and CDK2 inhibitors [112].

Similarly, we recently identified increased activation or overexpression of the protein synthesis pathway in the progression and resistance of highly aggressive MYC-amplified forms of MB [113]. Using genetic and pharmacologic approaches, we found that combined inhibition of MYC transcription (by BET protein inhibition) and mTOR signaling had a synergistic anti-tumor effect against MYC-driven group 3 MB [113]. Therefore, the MYC/mTOR axis could be an attractive therapeutic target for group 3 MB. Other important metabolically active druggable pathways in group 3 MBs include the folate and purine metabolism pathways [114], angiogenesis pathways [115], and CD47 [116]. According to these studies, the combined application of the folate synthesis inhibitor pemetrexed and nucleoside analog gemcitabine inhibited tumor cell growth [114], as did the humanized anti-CD47 antibody (Hu5F9-G4) [116].

In summary, most preclinical in vitro and murine models resemble MYC-activated MBs, and the field lacks adequate representation of heterogeneity within group 3 tumors. In fact, all existing group 3 MB cell lines are MYC-amplified. Model systems exploring tumorigenic mechanisms of non-MYC-amplified group 3 are needed. Most preclinical therapeutic studies focus on the inhibition of PI3K and mTOR signaling pathways, evaluate the synergistic activity between histone deacetylase inhibitors and PI3K inhibitors, assess the efficacy of CDK or BET bromodomain inhibitors, and test the effectiveness of anti-vascularization therapies. Such experiments are no longer confined to preclinical model systems, and numerous early phase clinical trials have started to explore these promising avenues in recurrent/refractory MBs.

## 5. Group 4 Medulloblastoma

Group 4 MB is the most-prevalent subgroup, accounting for almost 40% of all MB tumors, frequently seen in children aged 5–10 years and rarely in infants [22,38]. Arising within the fourth ventricle from unipolar brush cells, group 4 tumors occur three times more often in males than females (Figure 1) [13,91,92,117]. Thus, this subgroup has the most skewed distribution amongst all the subgroups. Similar to group 3, the classic and LCA histology are prevalent in these tumors. Patients with average-risk, as determined by the absence of metastasis, have a 5-year overall survival >80%. However, for the approximately 35–40% of patients exhibiting metastatic disease at diagnosis (making them high risk), prognosis is poor. Overall, patients with group 4 MB have an intermediate prognosis when treated with standard therapy (Figure 2) [7,89,100]. 

### 5.1. Molecular Characteristics and Subtypes of Group 4 MB

Although group 4 MB is most common in incidence, similar to group 3 tumors, underlying genetic or molecular basis for this disease is not well understood. Gross chromosomal gain (7 and 18q) and losses (8p, 10q and 17p) are frequently observed in group 4, particularly isochromosome 17q (i17q) present in ~80% of cases. Aberrant 11p and 18q and complete loss of X chromosomes (in females) are frequently observed in group 4 [13,18,93,98,118].

Frequent genetic markers mutated or altered in copy numbers include histone lysine demethylase *KDM6* (mutation), histone methyltransferases *KMT2D* and *KMT2C* (mutation), *OTX2* (amplifications), *DDX31* (deletions), *CHD7* (mutations), and activation of GFI1/GFI1B expression. These genetic alterations significantly overlap with those associated with group 3 MB. However, unlike group 3 tumors where *MYC* gene is amplified, preferential amplification of *MYCN* is observed in group 4 tumors [7,62,96,119]. In fact, a certain level of MYCN expression might be required for tumor maintenance. For example, when conditional MYCN^T58A^ (stabilized form of MYCN) was induced in neural cells in mouse cerebella from postnatal day 1 through adulthood, mice developed group 4-like tumors [120].

The most frequent genetic mutation seen in group 4 MB occurs in the histone demethylase gene *KDM6A*, which regulates the H3K27 methylation. As an epigenetic regulator, H3K27 regulates expression of genes that are involved in cell cycle control, differentiation, and pluripotency. Therefore, by preventing H3K27 demethylation, *KDM6A* mutation might preserve or initiate stem cell-like states in tumor cells. Interestingly, *KDM6A* seems to be more frequently mutated in boys with MB compared with girls, which might explain the observed male predominance of this MB subtype [19,121,122]. Another prevalent event in group 4 tumors is *CDK6* amplification and overexpression of PRDM6 via enhancer hijacking, which occurs along with *SNCAIP* duplication [62,93,119].

These mutational analyses along with integrated gene expression and DNA methylation arrays have identified the heterogeneity within group 4 tumors, giving rise to three primary subtypes α, β, and γ (Table 1). Group 4α are enriched for *MYCN* amplifications, whereas β are strongly enriched for *SNCAIP* duplications and presence of i17q. Group 4α and 4γ are strongly enriched for 8p loss and 7q gain. In addition, groups 4α and 4γ are enriched for focal *CDK6* amplifications [22].

### 5.2. Pre-Clinical Group 4 Models Identify Druggable Pathways

Current standard therapy cures a high proportion of patients with average-risk group 4 MB [68]. A prospective study of average-risk group 4 patients aged 3–17 years treated with surgery, irradiation, and chemotherapy found excellent 5-year progression-free survival (95.9% and 88.7%) for patients treated by two different protocols [123]. However, there are no therapies for high risk metastatic group 4 MBs, although upfront metastatic disease are somewhat salvageable with current treatment strategies [27,29]. As metastatic MB is the key risk factor for treatment failure in group 4 MB, inclusion of novel agents to the standard first-line treatment need to be considered. Preclinical studies have shown involvement of P13K signaling pathway as a potential driver pathway for recurrent, metastatic MB [97]. Further, absence of a mouse model that mimics human group 4 MB has hampered the identification of novel target or development of new approaches to treat patients with high-risk diseases. However, a recent study showed that group 4 tumors displayed aberrant receptor tyrosine kinase (RTK) ERBB4-SRC signaling. Enforced in utero expression of an activated SRC combined with *TP53* inactivation induced murine tumors that resembled group 4 MB [124]. These highly distinct RTK activity profiles distinguishing group 3 and 4 tumors were cross-validated in a non-overlapping, independent mass spectrometry dataset analysis [125]. This model suggested that group 4 MBs could be susceptible to available kinase inhibitors including lapatinib or dasatinib to improve outcomes in children with high-risk group 4 tumors.

## 6. Epigenetic Drivers of MB

Next-generation sequencing of MB tumors have shown that besides amplification of *MYC, MYCN, CCND2*, and *GLI2,* mutations in epigenetic regulator, e.g., histone lysine methyltransferase or acetyltransferase (KMT and HAT, also called “writers”); demethylases or deacetylases (KDM, HDAC, also called “erasers”); and members of the polycomb transcriptional repressor complex, PRC2 and PRC1, account for majority of the genetic perturbations in the non-SHH/WNT groups 3 and 4 MBs (Figure 3) [19,122,126]. The highest frequency of mutations in histone modifier was found in KDM6A, the H3K27 demethylase described earlier. KDM6A associates with H3-K4 methyltransferases, MLL2 and MLL3, which are also recurrently mutated along with *CHD7*. Consistent with this, aberrant patters of H3K4 and H3K27 histone methylation have been found in group 3 and 4 tumors. These altered epigenetic marks are thought to drive cell proliferation and a stem cell-like phenotype in MB [127].

The polycomb repressor complex PRC2 plays an important role in differentiation and maintenance of cell identity. EZH2, the catalytic subunit of the PRC2, is responsible for methylating the lysine 27 of H3, which contributes to chromatin compaction and transcriptional repression [128]. Studies have shown that a subset of group 3 and 4 tumors, but not SHH and WNT tumors, have high levels of EZH2 expression and H3K27 Me3 marks but low H3K4 methylation, a combination of epigenetic marks consistent with a stem/progenitor cell like identity [127]. Although *EZH2* was thought to be an oncogene, deletion of *EZH2* in mouse and human group 3 cells by CRISPR/cas9 gene editing approaches have resulted in loss of PRC2 complex, accelerating tumor growth [129]. In contrast, in vitro studies in MB cell lines using shRNA or the EZH2 inhibitor suggested that EZH2 had oncogenic activity [130]. Therefore, EZH2 can function as a tumor suppressor or an oncogene in a context-dependent manner. Inactivation of EZH2 was found to accelerate MYC-driven group 3 MB by upregulation of *GFI1* proto-oncogene. Interestingly, GFI1 is a transcriptional repressor that is upregulated in group 3 MB by enhancer high jacking. GFI1 disruption antagonized the tumor-promoting effects of EZH2 loss; conversely, GFI1 overexpression worked synergistically with MYC to bypass effects of *TP53* inactivation in driving MB progression in primary cerebellar neuronal progenitors. Therefore, EZH2, MYC, and GFI1 may combine to drive group 3 MB through concerted mechanisms [96,129].

Another class of epigenetic regulators, BET-containing proteins, binds to acetylated histones in the promoter regions and in turn facilitates recruitment of positive transcription elongation complex-b (P-TEFB), comprising CDK9 and Cyclin T1, to activate transcription elongation [131]. Our recent study showed that CDK9 and BRD4 are upregulated in MB cell lines and in MB mouse model, and targeting CDK9 and BRD4 with small molecule inhibitors significantly decreased cell growth, migration, and gene expression [132]. BRD/BET proteins regulate the expression of the key driver oncogene *MYC* in group 3 MBs. Several studies have shown that BET inhibitor JQ1 can potently inhibit MB growth in a preclinical model, being currently under clinical trial for many other types of cancers [19,110,111,133,134].

Recurrent mutations in ATP-dependent chromatin remodeling SWI/SNF complex member *SMARCA4* have been identified in group 3 subgroups [19,135]. SWI/SNF and PRC2 complex exhibit epigenetic antagonism, raising the possibility that PRC2 inhibitors may work well in SWI/SNF mutant MBs [136]. Inactivation of ZMYM3, a histone-binding protein that contributes to gene transcription at the H3K4Me3 mark, occurs exclusively in group 4 MBs. *ZMYM3* mutations are often found along with *KDM6* mutations and EZH2 expression, suggesting the importance of H3K27/H3K4 modulation in MB [19,137]. 

## 7. Drug Resistance and Recurrence in MBs

A range of cellular mechanisms, including active efflux of the chemotherapeutic agents and efficient repair of DNA damage by multiple DNA repair pathways, have been proposed to induce chemo- and radio-resistance in tumor cells [138,139]. APE1, a key enzyme responsible for repair of radiation-induced oxidized bases or single-strand breaks via base excision repair pathway, is overexpressed in MB, and its activity is associated with response to radiation and cisplatin in MB patients [140,141]. Our recent study demonstrated that APE1 interacts with a chromatin remodeling histone chaperone Facilitates Chromatin Transcription (FACT) complex to facilitate radiation and cisplatin-induced DNA repair in group 3 MB cells in vitro. Notably, MB tissues have elevated levels of post-transitionally modified acetylated APE1 (AcAPE1) and FACT complex, both associated with poor prognosis [142]. Inhibition of FACT with small-molecule CBL0137 (also known as curaxin) sensitized MB cell lines and orthotopic tumors to radiation and cisplatin [142,143]. Curaxin modulates an important pathway for cancer stem cell proliferation, i.e., NOTCH signaling, [144,145]. Recent transcriptome analyses in group 3 MB cells showed that curaxin preferentially suppressed cell cycle and DNA repair-related biological processes. Treatment with curaxin downregulated the expression of hundreds of genes, including *MYC* and *OTX* genes that are known to be major activators of highly aggressive MB [142]. Further investigations as to whether curaxin can improve the efficacy of interventions for MB and outcomes in high-risk patients is warranted.

Repair of radiation- and cisplatin-induced DNA double-strand break (DSB) repair are critical to cell survival. DSBs are primarily repaired through homologous recombination (HR) or non-homologous end joining (NHEJ) pathways [146,147]. Therefore, deregulation of proteins or cofactors that control these repair pathways can promote DNA repair and induce drug resistance in cancer. PRKDC is overexpressed in MYC-amplified group 3 cell lines [97,125]. The *PRKDC* gene encodes the DNA-dependent protein kinase catalytic subunit (DNA-PKcs), which promotes DSBs via NHEJ [148]. The PRKDC inhibitor NU7441 preferentially sensitized MYC-amplified cells to radiation [97]. Similarly, growing evidence supports the importance of arginine methylation by protein arginine methyltransferases (PRMTs) in promoting drug resistance via modulating both HR and NHEJ. Specifically, PRMT5 activity was linked with NHEJ repair by methylating and stabilizing p53-binding protein 1, which facilitates NHEJ and promotes cellular survival after DNA damage [149]. PRMT5 is also a regulator of HR-mediated DSB repair, which is mediated through its ability to methylate RUVBL1, a cofactor of the TIP60 complex involved in HR [150]. Our study has demonstrated the overexpression of PRMT5 in MYC-driven primary MB tumors and cell lines compared to non-MYC MB tumors [151]. High expression of PRMT5 was shown to inversely correlate with patient survival. PRMT5 inhibitor EPZ015666 exhibited dose-dependent efficacy in suppressing cell growth and inducing apoptosis in MYC-driven MB [151]. There is evidence that PRMT5 cooperates with pICln to function as a master epigenetic activator of DNA damage response genes (*RAD51*, *BRCA1*, and *BRCA2*) involved in HR and NHEJ [152]. Taken together, these studies have revealed PRKDC and PRMT5 in promoting drug resistance via modulating DSBs repair.

## 8. Non-Coding RNAs in MB

Approximately 2% of transcribed RNA encodes proteins [153]. The remaining vast majority are non-coding RNAs (ncRNA), encompassing microRNAs (miRNAs), long non-coding RNA (lncRNAs), and circular RNAs (circRNAs). Despite differences in size, structure, and biogenesis, they are integral to nearly every step of gene expression, imposing regulatory control over epigenetic modifications, transcription, splicing, RNA stability, and translation [154,155]. In this section, we review the important diagnostic, prognostic, and pathologic roles of these ncRNAs in MB. 

### 8.1. MiRNA

MicroRNAs (miRNAs) are approximately 20–22 nucleotides in length and bind to the 3′UTR of target mRNA to regulate their expression [156]. MiRs possess the capability to both trigger and suppress growth on the basis of the rich network of target proteins affected by their binding. Thus, miRNAs can act as oncogenes or tumor suppressors depending up on their targets and the cellular context [157]. Deregulation of miRs is now considered a hallmark of multiple cancers and unique signatures for various miRs are currently being generated to guide cancer diagnostics and therapies. To define the role of miRs in medulloblastoma, researchers have conducted several studies, with some exploring the status of miRs specifically on chromosome 17p. 

In 2008, Ferretti et al. showed for the first time that Hh pathway regulates expression of miRs in MB. More specifically, they showed that miR-125b, miR-324-5p, and miR-326 target SMO and GLI1 [158]. In turn, overexpression of these miRs inhibited tumorigenesis by inhibiting Hh signaling. They further mapped the locus of miR-324-5p to 17p13.1, showing loss in 40% of human MBs [158]. A follow-up publication profiled 248 miRs in human MBs, revealing a total of 78 miRs to be differentially regulated in MBs, with miR-9 and miR-125a demonstrated as tumor suppressor genes [159]. We showed that activated STAT3 transcription factor mediates upregulation of miR-21 that suppressed protein inhibitor of activated STAT3 (PIAS3), thereby accelerating MB pathogenesis [160,161]. Moreover, miR-10b, miR-193a, miR-224-452 cluster, miR-182-183-96 cluster, miR148a, miR-23b, and miR-365 were specifically found in the WNT-MB subgroup [162,163]. Overexpression of miR-17-72 cluster (oncomiR) was frequently associated with SHH-MB [164]. Similarly, 12 miRNAs, including miR-181a, miR-135b, and miR-660, were overexpressed in group 4 MB [165]. 

In group 3 MB, loss of heterozygosity at locus 17p has been reported in >50% of group 3 MB cases, recapitulated in a local cohort of MB patients (GSE 148390) [166,167,168,169,170,171,172]. Along locus 17p13.3 lie multiple tumor suppressor genes, including three micro-RNAs: miR-22, miR-212, and miR-1253. Aside from being downregulated in medulloblastoma, forced expression of miR-22 in MB cell lines resulted in reduced tumorigenesis by downregulating 3′-phosphoadenosine 5′-phosphosulfate transporters 1 (PAPST1) [159,173]. Similarly, we have uncovered transcriptional silencing of miR-212 via histone modification in group 3 MB [174]. Specifically, in HDMB03 and D425 cells with baseline-reduced miR-212-3p expression, we showed enriched methylated H3K27 and high EZH2 activity, which, as discussed prior, are epigenetic markers consistent with a stem/progenitor cell-like identity [127]. Consequently, treatment with pan-HDAC inhibitors or EZH2 silencing returned miR-212 expressions to baseline. Induction of miR-212 within orthotopic tumors led to a significant augmentation of longevity in animal models concurrent with smaller tumor sizes. We also identified an important oncogenic target of miR-212, Nuclear Factor I/B, a target of c-Myc whose amplification in group 3 tumors is a cardinal high-risk feature [174]. Finally, on the terminal end of this locus lies miR-1253. In studying this micro-RNA, we revealed silencing of miR-1253 via hypermethylation in group 3 and 4 tumors [172]. On top of negatively affecting tumor cell viability, migration/invasion, and colony formation, miR-1253 activated apoptosis and led to cell cycle arrest at the G_0_/G_1_ phase. Oncogenic targets identified included CDK6, a cell cycle checkpoint marker for proliferation overexpressed in group 3 and 4 tumors [22], and CD279 (B7-H3), an immune checkpoint molecule that inhibits T-cell function and facilitates tumor proliferation and invasion [175]. 

### 8.2. LncRNA

LncRNAs can span over 200 nucleotides in length and are mostly transcribed by RNA polymerase II. Aside from their important role in the transcriptional, post-transcriptional, and epigenetic regulation of gene expression, they can play key roles in cell proliferation, apoptosis, migration, and invasion [176]. Deregulated expression of lncRNAs is associated with tumorigenesis of various cancers, including MB [155,177]. For example, lncRNA TP73-AS1 promoted the survival, migration, and proliferation of MB cells in vitro and in vivo by sequestering miR-494-3p and upregulating EIF5A2 expression [178]. In SHH MB, lncRNA NKX2-AS1 sequestered miR-103, miR-10, and miR-548m, downregulating essential tumor suppressors BTG2/LAST, and contributing to tumorigenesis [179]. Lnc-HLX-2–7 was highly upregulated in group 3 MB; silencing its expression significantly suppressed cell growth and induced tumor cellular apoptosis [180]. Linc-NeD125, which is significantly upregulated in group 4 MB, sequestered miR-19a-3p, miR-19b-3p, and miR-106a-5p, de-repressing key driver genes *CDK6*, *MYCN*, *SNCAIP*, and *KDM6A* [181]. 

Using high-throughput techniques, putative lncRNA signatures have been generated to differentiate medulloblastoma subgroups accurately [180,182]. One of these studies validated the top 10 oncogenic or tumor suppressive lncRNAs in each MB subgroup and explored their associated molecular pathways. Among these identified lncRNAs, high expression of HAND2-AS1 in WNT-MB, low expression of MEG3 in SHH-MB, high expression of DLEU2 and DSCR8 in group 3 MB, and high expression of DLEU2 and low expression of XIST in group 4 were associated with poor prognosis [183]. In this manner, lncRNAs may be exploited as prognostic biomarkers for molecular identification of MB subgroups.

### 8.3. CircRNA

Circular RNAs (circRNAs) are generated from specific splicing events where the 5′ and 3′ termini are covalently linked [184]. Possessing both tissue-specific and cell-specific expression patterns, they can function as miRNA sponges or protein decoys to influence various biological processes [184]. Their importance to MB pathophysiology is an emerging topic of high interest. Next-generation sequencing has identified 33 differentially expressed circRNAs in MB tissues. Of these, circ-SKA3 and circ-DTL were upregulated, while circ-CRTAM, circ-MAP3K5, circ-RIMS1-1, and circ-FLT3-1 were significantly downregulated in MB tissues. Downregulation of circ-SKA3 and circ-DTL suppressed MB cell proliferation, migration, and invasion by regulating the expression of host genes [185].

In summary, although the functional significance of the vast majority of ncRNAs remains undefined, mounting evidence on the role of ncRNAs in cancer tumorigenesis suggests a pivotal role in MB pathophysiology. Possessing subgroup specificity, ncRNA signatures may be exploited to personalize therapies by acting dually as biomarkers and therapeutic agents. 

## 9. Future Directions and Conclusions

Patients with medulloblastoma have been classically stratified into average-risk or high-risk groups [21,41]. More recent genome-based, high-throughput analytic techniques have helped subdivide medulloblastoma into four primary subgroups [100]. Coupling molecular, cytogenetic, and clinical features with methylation profiling and prognostic data has led to an even more refined subgroup classification into 12 subtypes [22]. However, despite advances in diagnostic accuracy, a large prognostic discordance exists between average risk (WNT and SHH) and high-risk (groups 3 and 4) tumors that continue to be treated similarly, widening the survival gap amongst patients [41]. Thus, our primary objective must be to leverage our advanced understanding of key signaling pathways to improve upon and tailor future therapies to the individual patient, capitalizing on the rich diagnostic accuracy we currently possess. Moreover, focus must shift towards slowly eradicating cycling tumor-initiating cells that are resistant to cytotoxic drugs and inhibitors, such as those expressing SOX2^+^ or high H3K27 methylation and EZH2 expression, and can eventually give rise to recurrent tumors recalcitrant to treatment. Targeted (individualized) therapies and drug combinations based on genetic and epigenic signatures will likely provide better outcomes with far less adverse effects and might be better tolerated by our youngest patients. Finally, many of the studies described in this review are based primarily on cell culture models that do not recapitulate the complexity of in vivo tumors. Thus, the field would benefit from improved pre-clinical and animal models, especially for group 3 and 4 MB, to help advance our understanding of the molecular basis for high-risk disease and to direct our efforts in generating and testing inhibitors that may squelch the gap in survival punctuating high-risk tumor subgroups. 

## Figures and Tables

**Figure 1 diagnostics-12-00061-f001:**
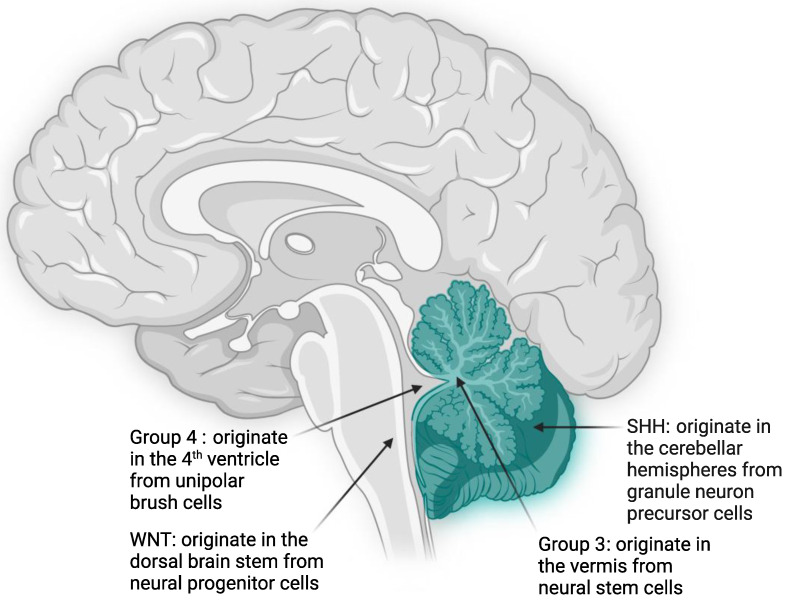
Origin of medulloblastoma tumors. Schematic showing the subgroup-specific origination and precursor cell for medulloblastoma tumors. Cerebellum is depicted in teal color. WNT, wingless subgroup; SHH, sonic hedgehog subgroup.

**Figure 2 diagnostics-12-00061-f002:**
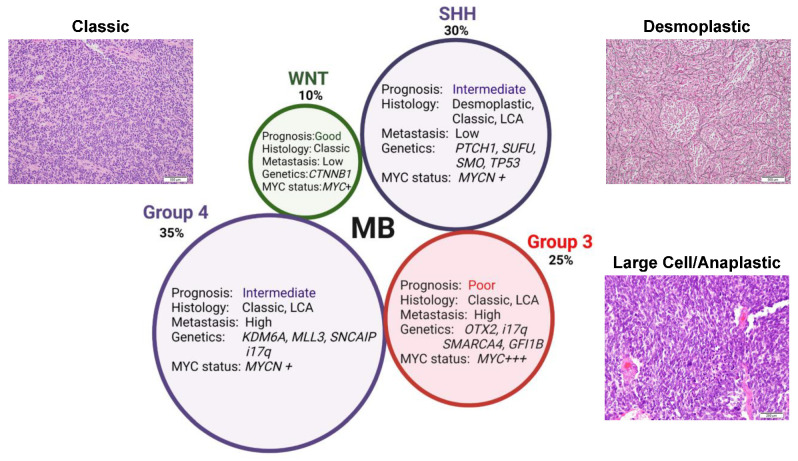
Medulloblastoma subgroup features and characteristics. Schematic depicting the four primary medulloblastoma subgroups and their unique features. Regarding histology, classic tumors are characterized by sheets of small, round, monomorphic cells possessing high nuclear-to-cytoplasmic ratios and hyperchromatic nuclei. Desmoplastic tumors are characterized by nodules of tumor cells displaying neurocytic differentiation within pale islands. Large cell/anaplastic tumors feature cells with nuclear enlargement, hyperchromatism, a high mitotic index, and atypical mitotic figures. Hematoxylin and eosin staining, 200–500X magnification. (Images courtesy of Deborah Perry, Children’s Hospital & Medical Center, NE). CTNNB1, catenin beta 1; GFI1B, growth factor 1B transcriptional repressor; i17q, isochromosome 17q; KDM6A, lysine demethylase 6A; LCA, large cell anaplastic; MLL3, mixed lineage leukemia protein 3; MYC, cytoplasmic MYC proto-oncogene; MYCN, nuclear MYC proto-oncogene; OTX2, orthodenticle homeobox 2; PTCH1, patched-1; SMARCA4, SWI/SNF-related, matrix-associated, actin-dependent regulator of chromatin, subfamily A member 4; *SMO*, smoothened; SNCAIP, synuclein alpha-interacting protein; SUFU, suppressor of fused; TP53, tumor protein 53.

**Figure 3 diagnostics-12-00061-f003:**
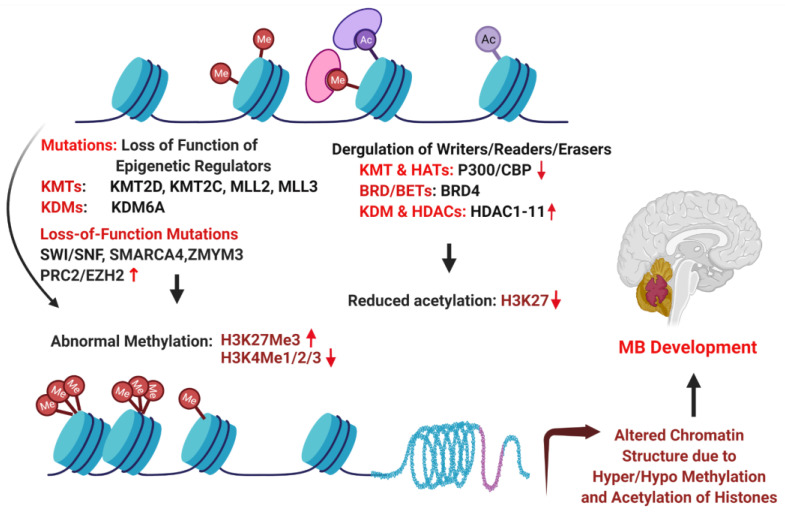
Epigenetic alterations in medulloblastoma. BRD4, bromodomain-containing 4; CBP, CREB-binding protein; EZH2, enhancer of zeste 2 polycomb repressive complex 2 subunit; H3K27Me3, tri-methylated lysine 27 on histone 3; H3K4Me, methylated lysine 4 on histone 3; HDAC, histone deacetylase; KDM6A, lysine demethylase 6A; KMT, lysine methyltransferase; MLL, mixed lineage leukemia protein; SMARCA4, SWI/SNF-related, matrix-associated, actin-dependent regulator of chromatin, subfamily A member 4; SWI/SNF, switch/sucrose non-fermentable chromatin remodeling complex; ZMYM3, zing finger MYM-type-containing 3.

**Table 1 diagnostics-12-00061-t001:** Tumor subtypes within medulloblastoma subgroups *.

Subgroup	Subtype	Frequency	Median Age (yrs)	Distinguishing Genetic Events	Metastasis Incidence	5-Year Overall Survival
** WNT **	WNTα	70%	10	>95% monosomy chromosome 6	9%	97%
	WNTβ	30%	20	~30% monosomy chromosome 6	21%	100%
** SHH **	SHHα	29%	8	Loss of 9q, 10q, 17p; gain of 9p; enriched in *MYCN*, *GLI2* amp and *TP53* mutations (35%)	20%	70% **
	SHHβ	16%	1.9	Significant gain in chromosome 2; focal *PTEN* deletions (25%)	33%	67%
	SHHγ	21%	1.3	Low copy number alterations	9%	88%
	SHHδ	34%	26	Enriched in *TERT* mutations (90%)	9%	89%
** Group 3 **	Grp 3α	46%	4.8	i17q; loss of 8q and 17p	43%	66%
	Grp 3β	26%	7.6	*OTX2* gain and *DDX31* loss; activation of *GFI1* and *GFI1B* oncogenes	20%	56%
	Grp 3γ	28%	5	i17q; 8q gain and *MYC* amp	39%	42%
** Group 4 **	Grp 4α	30%	8.2	i17q; loss of 8p; 7q gain; *MYCN* and *CDK6* amp	40%	67%
	Grp 4β	33%	10	i17q; 17p loss; *SNCAIP* dup	41%	75%
	Grp 4γ	37%	7	i17q; loss of 8p; 7q gain; *CDK6* amp	39%	83%

* Data compiled from Cavalli et al., *Cancer Cell 2017,* 31: 737–754 [22]. ** Five-year OS for SHHα decreases to ~30% with TP53 mutations. GFI1/1B, growth factor-independent; GLI2, effector of hedgehog signaling; i17q, isochromosome 17q; MYC, cytoplasmic MYC proto-oncogene; MYCN, nuclear MYC proto-oncogene; PTEN, phosphatase and tensin homolog; SNCAIP, synuclein alpha-interacting protein; TERT, telomerase.

**Table 2 diagnostics-12-00061-t002:** Patient risk stratification.

	Average Risk	High Risk
** Age at diagnosis **	≥3 years old	<3 years old
** Extend of post-operative disease (by MRI) **	≤1.5 cm	>1.5 cm
** Presence of metastasis **	No	Yes
** 5-year event-free survival **	85%	60–70%
** Intensity of craniospinal irradiation **	23.4 Gy	36–39 Gy *
** Adjuvant chemotherapy **	cisplatin, vincristine, cyclophosphamide or lomustine	cisplatin, vincristine, cyclophosphamide, and lomustine

* Children under age 3 are treated with radiation-sparing or delayed radiation therapy due to the devastating sequelae of whole neuroaxis radiation on neurocognitive development [31,32,33]; MRI, magnetic resonance imaging; Gy, grays.

**Table 3 diagnostics-12-00061-t003:** Current clinical trials in medulloblastoma.

Clinical Trial	Subgroups	Interventions	Phase
** NCT01878617 **	All	WNT: low dose CSI and lower dose cyclophosphamide SHH: vismodegib after standard chemotherapy G3/G4: adding permetrexed + gemcitabine to standard chemotherapy G3/G4: reduced dose cyclophosphamide	II
** NCT02066220 **	WNT	Low-risk: radiotherapy without carboplatin and reduced-intensity maintenance chemotherapy Standard-risk: radiotherapy with carboplatin and maintenance chemotherapy	II/III
** NCT02359565 **	Recurrent or Refractory MB	Pembrolizumab (MK-3475) every 21 days for 34 cycles	I
** NCT02644460 **	Recurrent or Refractory MB	Abemaciclib twice daily for 28 days with dose escalation for up to 2 years	I
** NCT02724579 **	WNT	Reduced CSI and no vincristine during radiotherapy followed by reduced-dose maintenance chemotherapy	II
** NCT03130959 **	Recurrent or Refractory MB	A: Nivolumab B: Nivolumab + Ipilimumab	II
** NCT03155620 **	Recurrent or Refractory MB	Targeted therapy based on genetic testing	II
** NCT03173950 **	Recurrent	Nivolumab weekly for 16 doses	II
** NCT03213678 **	Recurrent MB	Samotolisib twice daily for 28 days for up to 2 years	II
** NCT03434262 **	Recurrent or Refractory MB	SHH: Ribociclib + Sonidegib WNT/SHH: Ribociclib + Trametinib G3/G4: Ribociclib + gemcitabine	I
** NCT04023669 **	Recurrent or Refractory MB	All: Prexasertib + cyclophosphamide monthly up to 24 months G3/G4: Prexasertib + gemcitabine monthly up to 24 months	I

All data concerning clinical trials were obtained from ClinicalTrials.gov (accessed on 10 December 2021). CSI, craniospinal irradiation.

## Data Availability

Not applicable.

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
