# Peer review of "Subgroup-Specific Diagnostic, Prognostic, and Predictive Markers Influencing Pediatric Medulloblastoma Treatment"

_diagnostics, 2021, doi:10.3390/diagnostics12010061_

Round 1

Reviewer 1 Report

It is very comprehensive text, making opportunity for notice or comments.

Author Response

Thank you very much for your positive comments. 

Reviewer 2 Report

Dear co-authors,

you have written a very interesting review here. This will certainly add value to the current literature.
I would prefer more figures and tables to better illustrate their findings. Likewise, clinical illustrations (MRI) are also beneficial in a review.
To give the paper more expression, the adjuvant therapies should be elaborated further.

Author Response

You have written a very interesting review here. This will certainly add value to the current literature. I would prefer more figures and tables to better illustrate their findings. Likewise, clinical illustrations (MRI) are also beneficial in a review.

We thank the reviewer for their kind comments and suggestions. As per their recommendations, we have made the following modifications to the figures and tables: (i) Figure 1 is a new clinical illustration depicting the origins of each medulloblastoma subgroup. (ii) We have improved Figure 2 by adding H&E stains of the different types of tissue histologies found in each subgroup. (iii) Table 1 is new and provides details on the various subtypes that can be found within each subgroup, thus acknowledging the heterogeneity within subgroups. (iv) Table 2 is new and provides details on the current treatment-related risk stratification for medulloblastoma patients. (v) Table 3 has been improved by adding 3 additional active clinical trials testing immunotherapies in recurrent MB. Regarding MRI, we could not procure images from patients in the time frame for this revision given the need for ethical access and informed consent.

To give the paper more expression, the adjuvant therapies should be elaborated further.

We agree with the reviewer’s suggestion to elaborate adjuvant therapies in our review. In the revised introduction, we have detailed the various treatment regimens used, with specifics on craniospinal irradiation and adjuvant chemotherapies (also summarized in Table 2). We have added a paragraph on high-risk infant management with delayed irradiation, methotrexate, and autologous stem cell therapy (lines 112-118). In addition, we have discussed current immunotherapeutic  adjuvants in clinical trials for group 3 MB in section 4.2.

Reviewer 3 Report

In this review Ray et al describe the current genetic and epigenetic landscape of medulloblastomas and the potential avenues for new therapies using insights from different genomic and epigenomic insights.

General comments: The authors here review the extensive differences between molecular subgroups of medulloblastomas. The manuscript would benefit form a clearer and deeper understanding of the clinical aspects of treatment and prognosis currently known including the discussion of important recent clinical trials that show benefit in some of higher risk subgroups such as group 3 medulloblastomas.( Leary et. Al lancet 2021).

Specific comments:

  • Introduction line 2, cell of origin and location of medulloblastoma vary according to molecular subtype with evidence of it’s relationship to location with SHH tumors arising within the cerebellum, while wnt tumors arise from the lower rhombic lip. Group 3 and 4 seem to arise from the vicinity of the fourth ventricle. I suggest referring to 2010 Dec 23; 468(7327): 1095–1099.
  • line 43 - What is a high rate of metastasis here? With current treatments medulloblastomas have a survival rate ~70%, are there differences in prognosis in metastatic patients between molecular subgroups? Namely WNT and group 4 versus group 3?
  • line 45 - Please clarify The classification of medulloblastoma into different subgroups is based on molecular transcriptomic/methylation profiles. Although there are specific clinical pathological characteristics within these groups, these did not contribute to the classification as it reads here.
  • line 48 - The authors here refer to the classic classification of medulloblastomas into high and average risk which includes age and extent of disease as referenced in paragraph 2 line 6 (paper from 2004), however the molecular subgroups do not fall clearly into high vs average as suggested by the authors here. Please refer to the consensus stratification paper 2016: Acta Neuropathol (2016) 131:821–831
  • Line 53 – this statement is misleading, in that it implies that the classification of WNT and SHH drove improvement of treatment in these subgroups when in reality with classic surgical, radiotherapy and chemo regimens the field has identified subgroups with better prognosis namely WNT and infant SHH(non TP53 mutant).
  • Line 55 – this is another misconception as children below 3-5 are actually treated with high dose chemotherapy regimens and radiation sparing approaches. Therefore there are no age based limitation to chemotherapy, but rather cranial-spinal radiation therapy not used in this age cohort due to dismal cognitive morbidity among others.
  • Line 78-Metastatic WNT cases still have good prognosis when treated according to current standard of care. Refer to February 2014 JCO 32(9)
  • Line 123 – define the context of standard risk here as different studies will have different criteria here better defined as patients with WNT tumors
  • Line 202- In the scenario of infant SHH medulloblastoma it would add to this section a discussion on the recent published trial SJYC07 which showed indeed better PFS in the gama subgroup (Lancet Oncol. 2018 Jun; 19(6): 768–784.)
  • Line 211 here another misconception regarding current treatment startegies for infants, where radiation sparing approaches are currently in use rather than a delay in radiation treatment.
  • Line 293 – extremely old reference in regards to treatment, please update.
  • Here the authors introduce the consensus risk stratification. I would suggest clarifying the risk stratification scenario and it’s changes over time in the introduction of this review paper.
  • Line 247 – in the context of group 3 medulloblastoma and clinical trials it is essential to refer to the recent published trial where addition of carboplatin during radiation had a significant impact in PFS.
  • Line 449 - Although metastatic recurrence in group 4 medulos has a dismal prognosis, upfront metastatic disease, can be salvageable with current treatment strategies. Please refer to Leary et al, lancet 2021; Gajar et al JCO 2021

Author Response

General comments:

The authors here review the extensive differences between molecular subgroups of medulloblastomas. The manuscript would benefit from a clearer and deeper understanding of the clinical aspects of treatment and prognosis currently known including the discussion of important recent clinical trials that show benefit in some of higher risk subgroups such as group 3 medulloblastomas (Leary et. al, lancet 2021).

We agree with the reviewer’s suggestion. We have added 2 new paragraphs to the introduction that detail clinical aspects influencing treatment and prognosis with a deeper discussion on current risk stratification and how it influences post-surgical management (lines 96-118 and Table 2). We have also added a discussion on the clinical trial suggested by the reviewer in this section and in section 4.2.

Specific comments:

Introduction line 2, cell of origin and location of medulloblastoma vary according to molecular subtype with evidence of it’s relationship to location with SHH tumors arising within the cerebellum, while wnt tumors arise from the lower rhombic lip. Group 3 and 4 seem to arise from the vicinity of the fourth ventricle. I suggest referring to 2010 Dec 23; 468(7327): 1095–1099.

We thank the reviewer for this suggestion. To better explore and review the cell and location of origin for medulloblastoma tumors, we have included a new illustration, i.e. Figure 1. We have deleted the statement, “They commonly arise along the floor of the fourth ventricle and grow to occupy the entirety of this cavity, subsequently invading both the cerebellar vermis and the brainstem” and replaced it with, “Accounting for 40% of tumors arising in the posterior fossa, these tumors can grow rapidly and invade important structures triggering cerebellar dysfunction and disrupting cerebrospinal fluid circulation (Figure 1).” Moreover, in each subgroup section, we discuss the cell and location of origin with appropriate references cited:

WNT: These tumors typically arise in the midline away from the cerebellum in the dorsal brainstem from progenitor cells originating in the lower rhombic lip during cerebellar development (Figure 1).1

SHH: Sonic hedgehog medulloblastomas (SHH MB) comprise approximately 30% of all MB and arise in the cerebellum from granule neuron precursor cells due to constitutive activation of the sonic hedgehog signaling pathway (Figure 1).1,2

Group 3: Group 3 tumors represent roughly 25% of all MB cases and arise from nestin positive neural stem cells in the cerebellar vermis (Figure 1).2-4

Group 4: Arising within the fourth ventricle from unipolar brush cells, group 4 tumors occur three times more often in males than females (Figure 1).3-6

Line 43 - What is a high rate of metastasis here? With current treatments medulloblastomas have a survival rate ~70%, are there differences in prognosis in metastatic patients between molecular subgroups? Namely WNT and group 4 versus group 3?

The reviewer is correct. Metastasis vary between the different subgroups and more so within the various subtypes. We have summarized these data in Table 1.

Line 45 - Please clarify The classification of medulloblastoma into different subgroups is based on molecular transcriptomic/methylation profiles. Although there are specific clinical pathological characteristics within these groups, these did not contribute to the classification as it reads here.

We have followed the reviewer’s recommendation. The statement has been modified as follows, “Genome-wide high-throughput analyses of large cohorts of medulloblastoma patients have revealed four distinct molecular subgroups, each possessing unique genetic and epigenetic alterations.6-16 Medulloblastomas are now subdivided into wingless (WNT), sonic hedgehog (SHH), group 3, and group 4 tumors, with each subgroup possessing a distinct transcriptomic and methylation profiles, somatic genetic aberrations, demographic distributions, histologies, and clinical outcomes (Figure 2).17,18

Line 48 - The authors here refer to the classic classification of medulloblastomas into high and average risk which includes age and extent of disease as referenced in paragraph 2 line 6 (paper from 2004), however the molecular subgroups do not fall clearly into high vs average as suggested by the authors here. Please refer to the consensus stratification paper 2016: Acta Neuropathol (2016) 131:821–831.

The reviewer is correct. We wanted to contrast the two different ways tumors are classified, i.e. risk stratification for treatment purposes vs. molecular subgrouping for diagnostic purposes. To do so, we first discuss diagnostic subgrouping in paragraph 2, followed by risk stratification for treatment in paragraphs 3 and 4. We end paragraph 3 with a statement discussing the need for consensus between the two systems moving forward given the results of recent clinical trials.

Line 53 – this statement is misleading, in that it implies that the classification of WNT and SHH drove improvement of treatment in these subgroups when in reality with classic surgical, radiotherapy and chemo regimens the field has identified subgroups with better prognosis namely WNT and infant SHH(non TP53 mutant).

We have removed this statement and significantly revised the introduction to reflect more accurately differences between subgroups.

Line 55 – this is another misconception as children below 3-5 are actually treated with high dose chemotherapy regimens and radiation sparing approaches. Therefore there are no age based limitation to chemotherapy, but rather cranialspinal radiation therapy not used in this age cohort due to dismal cognitive morbidity among others.

We have removed this statement and discussed high-risk infant management in paragraph 4 of the introduction.

Line 78-Metastatic WNT cases still have good prognosis when treated according to current standard of care. Refer to February 2014 JCO 32(9).

We have modified this statement to read as follows, “Even patients presenting with metastasis in 5-10% of cases experience good prognosis (Table 1).19

Line 123 – define the context of standard risk here as different studies will have different criteria here better defined as patients with WNT tumors.

We have defined standard risk here as near-total resection, beta-catenin nuclear staining, and non-metastatic tumors.

Line 202- In the scenario of infant SHH medulloblastoma it would add to this section a discussion on the recent published trial SJYC07 which showed indeed better PFS in the gama subgroup (Lancet Oncol. 2018 Jun; 19(6): 768– 784.)

This was a helpful suggestion. We now comment on this trial in lines 334-337. Specifically: “Currently, there are several ongoing clinical trials for SHH MB (Table 3). In addition, a recent clinical trial (SJYC07) reported that a subset of SHH MB patients (SHH-II) based on their DNA methylation status exhibited improved progression-free survival (PFS).20

Line 211 here another misconception regarding current treatment startegies for infants, where radiation sparing approaches are currently in use rather than a delay in radiation treatment.

Line 293 – extremely old reference in regards to treatment, please update.

Thank you for pointing this out. We have replaced this reference with two more recent reference.  Specifically, Shih DJ, Northcott PA, Remke M, et al., Cytogenetic prognostication within medulloblastoma subgroups. J Clin Oncol 32:886-896, 2014. And Severini LL, Ghirga F, Bufalieri F, et al., The SHH/GLI signaling pathway: a therapeutic target for medulloblastoma. Expert Opinion on Therapeutic Targets, 24:1159-1181, 2020. The modified statement now reads, “Progress on either of these experimental approaches could help develop novel strategies to either delay or prevent SHH MB recurrence, which occurs with an incidence of ~25% and is nearly always fatal.21,22

Line 247 – in the context of group 3 medulloblastoma and clinical trials it is essential to refer to the recent published trial where addition of carboplatin during radiation had a significant impact in PFS.

As suggested by the reviewer, we have discussed the recently published clinical trial (by Leary SES et al, JAMA Oncology 2021) in both the introduction and in section 4.2 of the revised manuscript.

Line 449 - Although metastatic recurrence in group 4 medulos has a dismal prognosis, upfront metastatic disease, can be salvageable with current treatment strategies. Please refer to Leary et al, lancet 2021; Gajar et al JCO 2021.

As per the reviewer’s suggestion, we have included this statement, “However, there are no therapies for high risk metastatic group 4 MBs, although upfront metastatic disease are somewhat salvageable with current treatment strategies.23,24

References

  1. Gibson P, Tong Y, Robinson G, et al. Subtypes of medulloblastoma have distinct developmental origins. Nature. 2010; 468(7327):1095-1099.
  2. Juraschka K, Taylor MD. Medulloblastoma in the age of molecular subgroups: a review. J Neurosurg Pediatr. 2019; 24(4):353-363.
  3. Vladoiu MC, El-Hamamy I, Donovan LK, et al. Childhood cerebellar tumours mirror conserved fetal transcriptional programs. Nature. 2019; 572(7767):67-73.
  4. Perreault S, Ramaswamy V, Achrol AS, et al. MRI surrogates for molecular subgroups of medulloblastoma. AJNR Am J Neuroradiol. 2014; 35(7):1263-1269.
  5. Northcott PA, Korshunov A, Pfister SM, Taylor MD. The clinical implications of medulloblastoma subgroups. Nat Rev Neurol. 2012; 8(6):340-351.
  6. Kool M, Korshunov A, Remke M, et al. Molecular subgroups of medulloblastoma: an international meta-analysis of transcriptome, genetic aberrations, and clinical data of WNT, SHH, Group 3, and Group 4 medulloblastomas. Acta neuropathologica. 2012; 123(4):473-484.
  7. Louis DN, Perry A, Reifenberger G, et al. The 2016 World Health Organization Classification of Tumors of the Central Nervous System: a summary. Acta neuropathologica. 2016; 131(6):803-820.
  8. Fattet S, Haberler C, Legoix P, et al. Beta-catenin status in paediatric medulloblastomas: correlation of immunohistochemical expression with mutational status, genetic profiles, and clinical characteristics. The Journal of pathology. 2009; 218(1):86-94.
  9. Hovestadt V, Jones DT, Picelli S, et al. Decoding the regulatory landscape of medulloblastoma using DNA methylation sequencing. Nature. 2014; 510(7506):537-541.
  10. Kool M, Koster J, Bunt J, et al. Integrated genomics identifies five medulloblastoma subtypes with distinct genetic profiles, pathway signatures and clinicopathological features. PloS one. 2008; 3(8):e3088.
  11. Korshunov A, Remke M, Kool M, et al. Biological and clinical heterogeneity of MYCN-amplified medulloblastoma. Acta neuropathologica. 2012; 123(4):515-527.
  12. Northcott PA, Shih DJ, Peacock J, et al. Subgroup-specific structural variation across 1,000 medulloblastoma genomes. Nature. 2012; 488(7409):49-56.
  13. Park AK, Lee SJ, Phi JH, et al. Prognostic classification of pediatric medulloblastoma based on chromosome 17p loss, expression of MYCC and MYCN, and Wnt pathway activation. Neuro-oncology. 2012; 14(2):203-214.
  14. Remke M, Hielscher T, Korshunov A, et al. FSTL5 is a marker of poor prognosis in non-WNT/non-SHH medulloblastoma. Journal of clinical oncology : official journal of the American Society of Clinical Oncology. 2011; 29(29):3852-3861.
  15. Robinson G, Parker M, Kranenburg TA, et al. Novel mutations target distinct subgroups of medulloblastoma. Nature. 2012; 488(7409):43-48.
  16. Wang X, Dubuc AM, Ramaswamy V, et al. Medulloblastoma subgroups remain stable across primary and metastatic compartments. Acta neuropathologica. 2015; 129(3):449-457.
  17. Gottardo NG, Hansford JR, McGlade JP, et al. Medulloblastoma Down Under 2013: a report from the third annual meeting of the International Medulloblastoma Working Group. Acta neuropathologica. 2014; 127(2):189-201.
  18. Taylor MD, Northcott PA, Korshunov A, et al. Molecular subgroups of medulloblastoma: the current consensus. Acta neuropathologica. 2012; 123(4):465-472.
  19. Cavalli FMG, Remke M, Rampasek L, et al. Intertumoral Heterogeneity within Medulloblastoma Subgroups. Cancer cell. 2017; 31(6):737-754 e736.
  20. Robinson GW, Rudneva VA, Buchhalter I, et al. Risk-adapted therapy for young children with medulloblastoma (SJYC07): therapeutic and molecular outcomes from a multicentre, phase 2 trial. The Lancet Oncology. 2018; 19(6):768-784.
  21. Shih DJ, Northcott PA, Remke M, et al. Cytogenetic prognostication within medulloblastoma subgroups. Journal of clinical oncology : official journal of the American Society of Clinical Oncology. 2014; 32(9):886-896.
  22. Lospinoso Severini L, Ghirga F, Bufalieri F, Quaglio D, Infante P, Di Marcotullio L. The SHH/GLI signaling pathway: a therapeutic target for medulloblastoma. Expert Opin Ther Targets. 2020; 24(11):1159-1181.
  23. Leary SES, Packer RJ, Li Y, et al. Efficacy of Carboplatin and Isotretinoin in Children With High-risk Medulloblastoma: A Randomized Clinical Trial From the Children's Oncology Group. JAMA Oncol. 2021; 7(9):1313-1321.
  24. Gajjar A, Robinson GW, Smith KS, et al. Outcomes by Clinical and Molecular Features in Children With Medulloblastoma Treated With Risk-Adapted Therapy: Results of an International Phase III Trial (SJMB03). Journal of clinical oncology : official journal of the American Society of Clinical Oncology. 2021; 39(7):822-835.